Unique dental arrangement in a new species, Groenlandaspis howittensis (Placodermi, Arthrodira) from the Middle Devonian of Mount Howitt, Victoria, Australia.

Fitzpatrick Austin N. fitz0335@flinders.edu.au
http://orcid.org/0000-0003-0380-7347 Clement Alice M.
http://orcid.org/0000-0001-8012-0114 Long John A.
College of Science and Engineering, Flinders University of South Australia , Adelaide , Australia
Li Dongming
Electronic publication date: 2024 Dec 23
Publication date: 2024
Volume: 12
Electronic Location ID: e18759
Received 2024 Sep 20; Accepted 2024 Dec 4
Copyright: © 2024 Fitzpatrick et al.
Copyright year: 2024
Copyright holder: Fitzpatrick et al.
License: This is an open access article distributed under the terms of the Creative Commons Attribution License, which permits unrestricted use, distribution, reproduction and adaptation in any medium and for any purpose provided that it is properly attributed. For attribution, the original author(s), title, publication source (PeerJ) and either DOI or URL of the article must be cited.
License URL: https://creativecommons.org/licenses/by/4.0/

Keywords: Placoderm, Arthrodire, Devonian, Phylogenetic analysis, Dentition, Morphology

Funding: John Clema of New Mexico, USA This research was funded by a donation of funds to the research account of John A. Long by businessman John Clema of New Mexico, USA. The funders had no role in study design, data collection and analysis, decision to publish, or preparation of the manuscript.

==============================
Well-preserved specimens of a new species of arthrodiran placoderm, Groenlandaspis howittensis sp. nov. (Middle Devonian of Victoria, Australia), reveals previously unknown information on the dermal skeleton, body-shape and dentition of the wide-spread genus Groenlandaspis. The new material includes dual pineal plates, extrascapular plates, and cheek bones showing the presence of cutaneous sensory organs. The anterior supragnathal, usually a paired element in arthrodires, is a fused median bone in G. howittensis sp. nov. It is positioned anterior to the occlusion of the mouth between the lower jaw (infragnathals) and upper jaw (posterior supragnathals) bones, indicating a specialised feeding mechanism and broadening the known diversity of placoderm dental morphologies. G. howittensis sp. nov. differs from all other groenlandaspidids by a less pronounced posterior expansion of the nuchal plate; the shape of the posterior dorsolateral plate and the presence of a short accessory canal on the anterior dorsolateral plate. A new phylogenetic analysis positions Groenlandaspididae in a monophyly with the phlyctaeniid families Arctolepidae and Arctaspididae, however, the specific intrarelationships of groenlandaspidids remain poorly resolved.

Introduction

Arthrodires are an extinct clade of placoderms (the monophyletic status of placoderms is contentious see Brazeau, 2009, Brazeau & Friedman, 2014, also see Young et al., 2010 and King et al., 2016) and a dominant faunal component of Devonian marine and freshwater ecosystems. Arthrodires are one of the phylogenetically most basal jawed vertebrates to possibly show evidence of true teeth (Smith & Johanson, 2003; Rücklin et al., 2012; Rücklin & Donoghue, 2015; Vaškaninová et al., 2020, see Young, 2003 and Burrow, Hu & Young, 2016 for opposing interpretations) and provide valuable insight into the early evolution of feeding ecologies, including durophagy (Dennis & Miles, 1979), suspension feeding (Coatham et al., 2020) and pelagic hunting strategies (Jobbins et al., 2024). However, knowledge of these specialisations is generally limited to more derived forms, such as the Eubrachythoraci, which possess more robust jaw bones. Consequently, the dental morphology of more basal arthrodires, such as that of the globally occurring family Groenlandaspididae remain poorly understood. Groenlandaspidids are known from Lower to Upper Devonian deposits throughout Gondwana (Young, 1993; Anderson et al., 1999), attaining a cosmopolitan distribution following a northward dispersal into Laurussia in the Late Devonian (Janvier & Clément, 2005). The namesake genus, Groenlandaspis Heintz, 1932, was erected based on isolated head and trunk plates from the uppermost Devonian of Greenland (Stensiö Bjerg Formation) with further material, including the characteristic posterior dorsolateral plate, described by Stensiö (1934, 1936, 1939). The first Groenlandaspis plates were described by Woodward (1891) as “Coccosteus disjectus” from the Late Devonian Kiltorcan Formation of Ireland, many years later Ritchie (1975) would review the European material, reclassifying the Kiltorcan arthrodire as Groenlandaspis disjectus and describe a new species, Groenlandaspis antarcticus, from the Middle Devonian of Antarctica.

The genus Groenlandaspis, consists of 10 named species: G. disjectus, (Woodward, 1891), G. mirabilis Heintz, 1932, G. antarcticus Ritchie, 1975, G. seni Janvier & Ritchie, 1977, G. theroni (Chaloner et al., 1980; Anderson et al., 1999), G. riniensis Long et al., 1997, G. pennsylvanica Daeschler, Frumes & Mullison, 2003, G. thorezi Janvier & Clément, 2005, G. potyi Olive, Prestianni & Dupret, 2015, G. howittensis sp. nov. (this article), and numerous more occurrences categorized only to genus level (Young, 1993, Janvier & Clément, 2005, table 1). The Middle Devonian Mount Howitt fossil site (Victoria, Australia) preserves a diverse freshwater fish fauna (Table 1) as compressed articulated individuals displaying aspects of both dermal and visceral morphology (Long, 1983a, 1983b, 1984, 1986, 1987, 1988; 1992, 1999; Long & Werdelin, 1986; Long & Holland, 2008; Long & Clement, 2009; Holland, Long & Snitting, 2010). We herein describe well-preserved and extensive material of a new species, Groenlandaspis howittensis sp. nov., representing the first member of the globally-distributed genus to be formally described from Australia. This material reveals new features of the tooth plates, squamation and body-shape of the genus. Multiple characteristics have been suggested to be relevant to the evolution of groenlandaspidids (Long, 1995a; Olive, Prestianni & Dupret, 2015) but none have been incorporated into a computer driven analysis until now. New complete material such as this offers the opportunity to clarify the phylogenetic relationships of Groenlandaspis, and the family Groenlandaspididae.

Table 1 Faunal list from the Mount Howitt locality, Victoria, Australia following Long (1983a, 1999).

‘Placodermi’	
      Arthrodira	
            Phyllolepididae	
                   Austrophyllolepis ritchei, Long, 1984	
            Groenlandaspididae	
                   Groenlandaspis howittensis sp. nov.	
      Antiarchi	
            Bothriolepididae	
                  Bothriolepis gippslandiensis, Hills, 1929	
                  Bothriolepis cullodensis, Long, 1983a	
                  Bothriolepis fergusoni, Long, 1983a	
‘Acanthodii’	
     Climatiiformes	
           Culmacanthiidae	
                 Culmacanthus stewarti, Long, 1983b	
     Acanthodiformes	
                 Howittacanthidae	
                        Howittacanthus kentoni, Long & Werdelin, 1986	
Osteichthyes	
     Sarcopterygii	
           Coelacanthiformes	
                 Gavinia syntrips, Long, 1999	
           Dipnoi	
                 Howidipterus donnae, Long, 1992	
                 Barwickia downunda, Long, 1992	
           Canowindridae	
                 Beelarongia patrichae, Long, 1987	
           Tristichopteridae	
                 Marsdenichthys longioccipitus, Long, 1985	
           ?Elpistostegalia	
                 Howittichthys warranae, Long & Holland, 2008	
      Actinopterygii	
            Palaeonisciformes	
                Howqualepis rostridens, Long, 1988	

Materials and Methods

Fossil preparation. Specimens were collected from Taungurong country, Victoria, during field trips lead by Professor Jim Warren of Monash University between 1970–1974, and by the late Alex Ritchie of the Australian Museum in the early 1990’s. The Groenlandaspis material consists of specimens from the upper conglomerate and lower mudstone units of the Bindaree Formation (Long, 1983a). Specimens were prepared in 15% Hydrochloric acid (HCl) solution to dissolve friable bone to reveal both sides of an individual as impressions within the rock. Black latex casts were whitened with ammonium chloride to reveal fine anatomical detail for comparative analysis.

Phylogenetic analysis

We performed a phylogenetic analysis of selected phlyctaenioid arthrodires using a morphological character matrix modified from the matrix of 121 characters and 60 taxa of Zhu, Zhu & Wang (2015). Eleven new characters were identified from the literature or during the course of this research and incorporated in this existing matrix (Table 2, see Supplemental 4 for full character list), forming a new matrix of 132 characters and 72 taxa. The matrix was treated with MESQUITE v3.61 (Maddison & Maddison, 2019), some minor corrections were made (Supplemental 3). In addition to G. howittensis sp. nov. described herein, nine more taxa were added to the ingroup, including the type species for Groenlandaspis, G. mirabilis, Heintz, 1932 and four relatively complete groenlandaspidids: Tiaraspis subtilis (Gross, 1933), Groenlandaspis riniensis Long et al., 1997, and Africanaspis doryssa, Long et al., 1997, and Mulgaspis evansorum Ritchie, 2004, plus two arctolepidids (Arctolepis decipiens (Woodward, 1891), and Heintzosteus brevis (Heintz, 1929)). Two selenosteids, Alienacanthus malkowskii, Kulczycki, 1957 and Amazichthys trinajsticae Jobbins et al., 2022, were added for diversity. Using our modified matrix, a phylogenetic analysis was performed in PAUP* 4.0 (Swofford, 2003) using a heuristic search with a random addition sequence of 1,000 repetitions and holding 1,000 trees per search. Characters 4, 14, 20, 35, 51, 75, 92, 93, 126, and 128 were ordered as they form morphoclines. The tree was rooted using the actinolepid arthrodires Kujdanowniaspis podolica Stensiö, 1942 (retained from Zhu, Zhu & Wang (2015) ) and two additional taxa, Lehmanosteus hyperboreus, Goujet, 1984, and the genus Bryantolepis, scored as a composite of the species Bryantolepis brachycephela Camp, Welles & Green, 1949 and Bryantolepis williamsi Elliott & Carr, 2010. Outgroup taxa were selected for their completeness and sister relationship to Phlyctaenioidei, see the phylogenetic analyses of Dupret (2004) and Dupret, Zhu & Wang (2017).

Table 2 A total of 11 new characters added onto a matrix of 121 characters from Zhu, Zhu & Wang (2015).

No.	Description	Reference	
122	Cervical Joint: Sliding (0) Ginglymoid (1).	Miles (1973)	
123	Transversely divided pineal plate forming anterior and posterior plates: Absent (0) Present (1).	This article	
124	Cutaneous sensory pits present on the suborbital or/and post suborbital plates: Absent (0) Present (1).	King, Hu & Long (2018)	
125	Dermal contact between the anterior dorsolateral and posterior lateral plates: Absent (0) Present (1).	This article	
126	Inverted V-shaped flexure of the posterior dorsolateral plate sensory canal. Scored not applicable in taxa without a PDL sensory canal: No flexure (0) Weak flexure, >110° (1) Strong flexure, <110° (2).	Long (1995a)	
127	Dorsolateral ridge originating from near the condyle of the anterior dorsolateral plate: Absent (0) Present (1).	Long (1995a)	
128	Medial contact of the dorsolateral plates under the median dorsal plate: No contact (0) anterior dorsolateral plates (1) anterior and posterior dorsolateral plates (2).	Goujet (1984)	
129	Internal annular thickening of the posterior trunk plates (‘b.cpd’, Goujet, 1984, fig. 61B): Absent (0) Present (1).	Goujet (1984)	
130	Median contact of the posterior ventrolateral plate: Simple overlap (0) Sigmoidal/double overlapping (1)	Goujet (1984), Dupret (2004)	
131	Ventral sensory canals: Absent (0) Present (1)	This article	
132	Distinct infraspinal lamina/process (‘pr.infsp’, Miles & Westoll, 1968, fig. 40C; ‘la.spv’, Goujet, 1984, fig. 66A) of the anterior ventrolateral plate: Absent (0) Present (1).	This article	
133	Anterior ventral plates: Absent (0) Present (1)	Miles (1973)	

Systematic palaeontology

PLACODERMI McCoy, 1848

ARTHRODIRA Woodward, 1891

PHLYCTAENIOIDEI Miles, 1973

PHLYCTAENII Miles, 1973

GROENLANDASPIDIDAE Obruchev, 1964

GROENLANDASPIS Heintz, 1932

Type species: G. mirabilis Heintz, 1932

Amended Diagnosis. Groenlandaspidids with pineal element either singular or divided into dual anterior and posterior plates (APi and PPi); rostrally developed preorbital plates that contact the suborbital plate; postnasal plates absent. Extrascapular plates overlying a shallow posterior descending lamina. Anterior ventral plates absent. Large posterior dorsolateral plate with sharp V-shaped flexure of the lateral canal (<110°). Median dorsal plate longer than high.

Remarks. The generic diagnosis has not been updated since Stensiö (1939) described material of Groenlandaspis from East Greenland, then only consisting of the type species, G. mirabilis. Thereafter, additional species have been referred to the genus based on general resemblance, and researchers have since suggested that other groenlandaspidid genera, Africanaspis and Turrisaspis are nested within the genus and thus does not represent a monophyletic clade (Janvier & Clément, 2005; Olive, Prestianni & Dupret, 2015).

Groenlandaspis howittensis sp. nov.

Synonymy

“Groenlandaspis, sp.” (Long, 1983a, p. 297)

“Groenlandaspis sp.” (Long, 1984, p. 263)

“The gnathals of Groenlandaspis…” (Long, 1984, p. 294)

“the euarthrodire Groenlandaspis” (Long 1985, p. 1)

“Groenlandaspis sp.” (Long, 1992, p. 299)

“Groenlandaspis sp. nov.” (Young, 1993, p. 249)

“Groenlandaspis from Mount Howitt” (Long, 1995b, p. 119)

“specimens of an undescribed Groenlandaspis from Mt. Howitt, Victoria” (Long et al., 1997, p. 258)

“well-preserved adults of the Mt. Howitt Groenlandaspis” (Long et al., 1997, p. 258)

“one species of Groenlandaspis” (Long, 1999, p. 37)

“and groenlandaspidids, … with fused anterior supragnathals (based on Mount Howitt specimens, J.A. Long unpublished observations)” (Anderson et al., 1999, p. 161)

“Groenlandaspis sp.” (Burrow & Turner, 1999, p. 213)

“molds of articulated armours and tails of Groenlandaspis from the late Givetian?/ early Frasnian of Mt. Howitt, Victoria” (Burrow & Turner, 1999, p. 214)

“Groenlandaspis spp. … (Mount Howitt, Victoria, … Australia)” (Ritchie, 2004, p. 63)

“Groenlandaspis sp. nov. from Mount Howitt, Victoria, Australia” (Ritchie, 2004, p. 64)

“Groenlandaspis … from Mount Howitt” (Long, 2010, p. 85)

“Groenlandaspis_sp_Mt_Howitt” (King et al., 2016, p. 506)

“and in undescribed groenlandaspids, J. Long, pers. comm. March 2023” (Engelman, 2024, p. 28)

“Groenlandaspidae indet.” (Engelman, 2024, table 3, p. 38)

Diagnosis. Medium sized Groenlandaspis with an adult armour length up to 150 mm and a reconstructed total body length of approximately 300 mm. Skull-roof as long as broad with gently concaved posterior margin. Anterior dorsolateral plate possessing a short dorsal accessory canal. Posterior dorsolateral plate higher than long (NMV P48875, H/L = 1.44); lateral canal sharply flexed (between 96°, NMV P48875 and 105°, AMF 62437). Median dorsal plate sub-equilateral (H/L = approx. 0.65), caudal margin gently concave and lined with prominent tubercles.

Etymology. After the site where it was found at the base of Mount Howitt.

Holotype. NMV P48873, a complete specimen showing a flattened and complete headshield with partial lateral trunk shield and pectoral fin preserved (Figs. 1A, 1B). NMV P48874, counterpart to the holotype showing a complete ventral trunk shield (Figs. 1C, 1D) and gnathal plates (Fig. 2) preserved in life position.

Figure 1 G. howittensis sp. nov.

(A) Photo of the holotype NMV P48873, head shield and partial trunk shield in dorsal view. (C) Photo of NMV P48874, ventral trunk shield in ventral view. Latex peels whitened with ammonium chloride. (B, D) Sketch interpretations of same specimens.

Figure 2 G. howittensis sp. nov., NMV P48773, jaws in ventral view.

Latex peel whitened with ammonium chloride.

Referred Specimens. AMF 62532 (Figs. 3, 4), 63548 (Fig. 5), 62534 (Fig. 6), 62333 (Fig. 7), 155378 (Fig. 8), 63543 (Fig. 9), 62537 (Fig. 10), 62437. NMV P48875 (Fig. 11). P48884 (Fig. 12), P254749 (Figs. 13, 4).

Figure 3 G. howittensis sp. nov. head and trunk shield in dorsal view.

(A) Photo of AMF 62532, latex peel whitened with ammonium chloride. (B) Interpretive line drawing of same specimen, dotted lines indicate broken or incomplete plate margins.

Figure 4 G. howittensis sp. nov., pineal and cheek plates.

(A) Photo of the pineal plate of NMV P48873. (B) interpretive drawing of the same specimen. (C) Photo of the APi and PPi of AMF 62532. (D) Photo close-up of the cheek plates in NMV P48873. (E) Isolated suborbital plate of NMV P254749. (A, C, D, E) Latex peels whitened with ammonium chloride.

Figure 5 G. howittensis sp. nov., AMF 63548, skull roof in dorsal view.

(A) Latex peel whitened with ammonium chloride. (B) Interpretive line drawing of same specimen, dotted lines indicate unknown margins.

Figure 6 G. howittensis sp. nov., AMF 62534, juvenile head shield in ventral view.

Latex peel whitened with ammonium chloride.

Figure 7 G. howwitensis sp. nov., AMF 62333, gnathal plates in ventral view, latex peel whitened with ammonium chloride.

Figure 8 G. howittensis sp. nov., disarticulated head and trunk plates.

(A) Photo of AMF 155378, latex peel whitened with ammonium chloride. (B) Interpretive drawing of the same specimen, shaded areas indicate the internal side of the plate.

Figure 9 G. howittensis sp. nov., AMF 63543, partial ventral trunk shield in ventral view.

Figure 10 G. howittensis sp. nov., partial tail and lateral trunk plates in lateral view.

(A) Photo of AMF 62537 MD, PDL, ADL and tail depicted. (B) Closer view of the squamation, pelvic girdle and fins of the tail. (A, B) Latex peels whitened with ammonium chloride.

Figure 11 G. howittensis sp. nov., disarticulated trunk plates and tail in lateral view.

(A) Photo of NMV P48875, latex peel whitened with ammonium chloride. (B) interprative drawing of the same specimen, shaded areas indicate internal side of the plate.

Figure 12 G. howittensis sp. nov., in ventral view.

(A) Photo of NMV P48884, latex peel whitened with ammonium chloride. (B) interpretative drawing of the same specimen, shaded areas indicate internal side of the plate.

Figure 13 Sub-adult G. howittensis sp. nov., disarticulated trunk plates.

(A) Photo of NMV P254749, latex peel whitened with ammonium chloride. (B) Interprative drawing of the same specimen, shaded areas indicate internal side of plate.

Locality, Horizon, and Age. G. howittensis sp. nov. remains are known from the upper sandstone conglomerate and lower mudstone shale members of the Bindaree Formation exposed at the Mount Howitt Spur fossil site (Long, 1983a). The holotype is from the lower shale member. The age of the Mount Howitt fauna is considered to be Givetian based on evidence of its faunal composition and comparison with other Devonian fish faunas in south-eastern Australia (Young, 1993, 2007; Long, 1999; Long et al., 2021).

Results

Description

Skull roof. The skull roof of G. howittensis sp. nov. is known from complete and partial specimens (Figs. 1, 3, 5, 6). It is overall very similar to G. antarcticus (Ritchie, 1975, fig. 2A) but differs by its larger orbits and shape of the nuchal plate. The cranial sensory canals broadly adhere to the pattern described in other species of Groenlandaspis where complete crania are known, G. antarcticus and G. riniensis (Ritchie, 1975; Long et al., 1997). The anterior end of the supraorbital canal (soc) is typically deflected laterally (NMV P48874, Figs. 1A, 1B, AMF 63548, Fig. 5) but is deflected medially on the left preorbital plate in AMF 62532 (Fig. 3). As in other species of Groenlandaspis the preorbital lamina of the preorbital plate (PrO) is well developed and forms the preorbital corner of the orbit usually occupied by the postnasal plate in other phlyctaeniids, e.g., Dicksonosteus (Goujet, 1984, fig. 31) and the basal groenlandaspidid Mulgaspis (Ritchie, 2004, fig. 4E), we suspect they are completely reduced as in Arctolepis (Goujet, 1984). The pineal element of G. howittensis sp. nov. is formed of anterior (APi) and posterior pineal (PPi) plates, and in articulation they form approximately one third of the cranial length (Figs. 1A, 1B). The APi exhibits the pineal pit (ppt) on its dermal surface. In the holotype (NMV P48873) and one other specimen (AMF 63548) of G. howittensis sp. nov. the APi and PPi are fused and the suture is faint (Figs. 4A, 4B) but other specimens clearly show both plates in association but disarticulated e.g., AMF 62543 (Figs. 3 and 4C). Dual pineal plates are a distinct feature in some members of the Groenlandaspididae and, thus far, one or both plates have also been figured for Turrisaspis, Africanaspis, and Colombiaspis (Olive et al., 2015, 2019; Gess & Trinajstic, 2017). A dual pineal element is presumed to be present in Tiaraspis based on the gap in the headshield once reconstructed (Schultze, 1984, fig. 2), though the pineal figured by Schultze (1984) should be the APi given the presence of the pineal pit. Dual pineal plates are herein figured for the first time in a species of Groenlandaspis but have been previously noted, though not figured, in other species: G. disjectus, G. antarcticus and Groenlandaspis sp. from Canowindra, New South Wales, Australia (Ritchie, 2004, p. 63 and AN Fitzpatrick, 2024, personal observation) but are not confirmed for G. riniensis from the Waterloo Farm Lagerstatte, South Africa. The central plates are essentially the same as G. antarcticus, differing only in a deeper embayment area for the postorbital plate (PtO). The PrO embayment is shallowly developed in G. howittensis sp. nov. like G. antarcticus, except in the holotype NMV P8874, where it is straight (Fig. 1B). The nuchal (Nu) plate is longer than broad (B/L = 0.6, NMV 48874, Figs. 1A, 1B) and is roughly 40% of the cranial length; it is transversely convex, rising posteriorly to a slight median crest. The lateral margins of Nu are subparallel and do not posteriorly expand to the extent of other species, e.g., G. antarcticus (Ritchie, 1975, fig. 2A), G. riniensis (Long et al., 1997, fig. 10), G. thorezi (Janvier & Clément, 2005, fig. 1); in this aspect the Nu of G. howittensis is more similar to Mulgaspis (Ritchie, 2004, fig. 4E) and other phlyctaeniids such as Arctolepis (Goujet, 1984, fig. 77). The Nu plates posterior margin is bordered by small postnuchal processes of the paranuchal plates (PNu). Extrascapular plates (ESc) are preserved within the nuchal gap of one articulated specimen (Fig. 5) and a fragment of a possible dissociated ESc is also identified in AMF 155378 (Fig. 8). As in brachythoracids, e.g., Millerosteus minor (Desmond, 1974, fig. 1C), the extrascapulars are subtriangular paired plates which overlie the posterior descending lamina (pdl) of the skull-roof (Fig. 1B) and are furrowed by a sensory canal; unlike brachythoracids, this sensory canal does not converge with the occipital cross commissure (occ) of the PNu, instead arcing posteriorly, and may have superficially connected with the dorsal accessory canal (acc) of the ADL plate. The visceral surface of the skull-roof (Figs. 6, 12) displays no continuous nuchal or occipital thickening as developed in brachythoracids, though infranuchal pits (if.pt) are present, as in Parabuchanosteus (Young, 1979) and many other taxa.

Cheek plates. The cheek unit comprises large submarginal (SM) and suborbital plates (SO) divided by a slender post suborbital plate (PSO). All bones of the cheek are preserved in the holotype though the SM and PSO are crushed (Figs. 4E, 4D); the left SM and the right SO are preserved in AMF 62532 (Fig. 3); internal side of the SO and SM can be identified in one juvenile specimen AMF 62534 (Fig. 6). The SO is short and deep the suborbital lamina which formsthe ventral portion of the orbit contacts the PrO as in some eubrachythoracids, e.g., Eastmanosteus (Dennis-Bryan, 1987, fig. 5). The dermal surface of the plate carries two deep sensory lines, the supraoral (sorc) and infraorbital canals (ioc), which meet in the radiation centre of the plate (Fig. 4), the infraorbital and supraoral canals are discontinuous in NMV P48874 and NMV P254749 (Figs. 4E, 4D). All specimens where the SO is preserved exhibit a cutaneous sensory pit (cuso) just posterior to the radiation center (Figs. 3, 4E, 4D). The PSO is preserved in the holotype with the ventral portion of the plate broken and disarticulated (Fig. 1), it is a slender bone which tightly fits into the posterior notch of the SO plate. Its dermal surface is furrowed longitudinally by the postorbital sensory canal (psoc). The submarginal plate (SM) is preserved close to life position but broken in the holotype; in one specimen, AMF 62532 (Fig. 3), the SM is near complete with only the posterior margin obscured, it is displaced anterior to its life position and does not exhibit any of the sensory canals and thus cannot be the left SO plate. The SM of G. howittensis sp. nov. is the first of example of this bone described for a groenlandaspidid. It is a large, ellipsoidal bone which overlapped the lateral margin of the skull roof and postbranchial lamina of the AL plate, as in other basal arthrodiran forms, e.g., Wuttagoonaspis and Dicksonosteus (Ritchie, 1973; Goujet, 1984).

Gnathal plates and parasphenoid. The gnathal plates are preserved as impressions in NMV P48773, P48884 and AMF 62534, 62333 (Figs. 2, 6, 7, 12), but are best represented in the counterpart of the holotype where the infragnathals (IG) are superimposed onto the posterior supragnathals (PSG) (Fig. 2). The crescentic denticulated bone positioned under the rostral plate in this specimen, and others (NMV P48884 and AMF 62534, 62333) is interpreted as a fused anterior supragnathal (ASG) derived from the ancestral paired condition of other arthrodires, e.g., Actinolepis (Mark-Kurik, 1985, fig. 3) and buchanosteids (Hu, Lu & Young, 2017, fig. 2B). In one smaller individual (AMF 62534) the ASG is much slenderer in proportions, suggesting positive allometric growth in this element through ontogeny (Fig. 6). The oral surface of the ASG is covered in densely packed, rounded denticles which radiate laterally from a medial point. The parasphenoid is preserved in two specimens, (Figs. 6, 7). In ventral aspect, it is a small denticulated bone, as in other groenlandaspidids, Turrisaspis elektor (Daeschler, Frumes & Mullison, 2003, fig. 8) and Mulgaspis evansorum (Ritchie, 2004, fig. 4F). However, it is not preserved sufficiently well to provide additional anatomical detail. Visible in the holotype (Fig. 2), scattered over the ventral surface of the IG and PrO plates, are small, crenulate scales with deep surface grooves. These were possibly scales covering the underside of the head.

The posterior supragnathals (PSG) are elongated, dorsoventrally flattened paired bones which almost meet on the midline, just anterior to the pineal pit on the visceral surface of the APi. The oral surface is slightly concave across its long axis and entirely covered in small, densely-packed, rounded denticles that radiate from a posteromedial depression, with the largest denticles occupying the outermost margins. The posterior supragnathals of G. howittensis sp. nov. are almost identical in structure and position to the “supragnathals” of T.elektor (Daeschler, Frumes & Mullison, 2003, fig. 8) and “anterior supragnathals” of A. doryssa (Gess & Trinajstic, 2017, fig. 2B). Therefore, these gnathal plates are presumed homologous with the PSGs of the Mount Howitt species.

The infragnathal (IG) is preserved in ventral aspect in NMV P48873 (Fig. 2), P48884 (Fig. 12), and AMF 62333 (Fig. 7) and the oral surface is preserved partially in NMV P48884 but best represented by one juvenile specimen, AMF 62534 (Fig. 6). Overall, the IG resembles that of the buchanosteid, ANU V244 (Hu, Lu & Young, 2017; Hu et al., 2019), it is a long and slender bone with a slight mesial curvature it is assembled of two equally developed medial and lateral laminas which form a deep meckelian groove (v.gr, Figs. 2, 7) which would have housed the dorsal edge of the meckelian cartilage in life. The oral surface of the IG is entirely covered by short, rounded, densely packed denticles, as in phyllolepidids (Long, 1984; Ritchie, 2005), thus precluding the abductor division or “non-biting portion”. Denticles on the IG vary in depth and circumference, two major divisions can be determined: larger denticles on the distal (mesial) half and smaller denticles on the proximal (lateral) half (Fig. 6). This suggests denticles radiate anteriorly and posteriorly along the occlusal margin from the ossification center in the middle of the plate, as in the buchanosteid, ANU V244 (Hu et al., 2019, fig. 2).

Trunk plates. The trunk armour consists of the same dermal plates as in other groenlandaspidids, e.g., G. antarcticus and G. pennsylvanica (Ritchie, 1975; Daeschler, Frumes & Mullison, 2003). Anterior ventral plates are absent. The posterior trunk shield exhibits a well- developed ‘annular bourrelet’, (‘b.cpd’, Goujet, 1984, fig. 61B) along the posterior complex of plates (PDL, PL and PVL, Figs. 8, 13) as in other phlyctaeniids, such as Dicksonosteus and Arctolepis. The anterior dorsolateral plate (ADL) is preserved in external view in the holotype, NMV 48873 (Figs. 1A, 1B), AMF 62532 (Fig. 3), and AMF 62537 (Fig. 10), midline contact of the ADLs can be observed in AMF 62532 (Fig. 3) and NMV P48884 (Fig. 12). A short dorsal accessory canal (acc, Fig. 1B) is observed in the holotype and AMF 62537 (Figs. 1, 10) however cannot be observed in AMF 62532 as the respective surface of the ADL is covered by matrix. A short accessory canal is a feature unique to G. howittensis sp. nov. within the genus, but also present in the Early-Middle Devonian groenlandaspidid Mulgaspis (Ritchie, 2004, figs. 7A, 13). The posterior dorsolateral (PDL) is higher than long and is best preserved in NMV P48875 (H/L = 1.44, Fig. 11). The plate displays the characteristic symphysial surface for the opposite PDL (symph.s, Fig. 8) and inverted V-shaped lateral line sensory canal, which are considered diagnostic for the genus (Daeschler, Frumes & Mullison, 2003, fig. 5, Janvier & Clément, 2005, fig. 8). The angle of the dorsal flexure of the lateral canal is measured from 96° (NMV P48875) to 105° (AMF 62437) in the examined material. This variability likely due to the angular shear of the Mount Howitt specimens (see Fig. 5, and described in Austrophyllolepis (Long, 1984). The posterior lateral overlap area (oa.PL) bears a deep groove which corresponds to the annular bourrelet (ab) crossing the internal surface of the posterior lateral plate (PL, Fig. 8). The tip of the MD is usually missing in large individuals e.g., AMF 62537 (Fig. 10) and NMV P48875 (Fig. 11). A complete MD of a subadult specimen is preserved in lateral aspect in NMV P254749 (Fig. 13) and thus presents the only complete example of an MD to take accurate lateral measurements from. In G. hoittensis sp. nov. the MD plate is approximately sub-equilateral in shape (H/L = 0.65, NMV P254749, Fig. 13), in all specimens its ventral margin is deeply scalloped and the ornamentation radiates from the dorsal apex of the plate developing into prominent tubercles along the caudal margin. The spinal plate (Sp) is identical to that of G. antarcticus, except for the variable presence of tiny hook-like spines on the mesial margin of the spinal plate (Figs. 1, 11, 12).

The ventral surface of the trunk shield is crushed but completely preserved in the holotype (Figs. 1C, 1D). The anterior median ventral plate (AMV) is broader than long (B/L = 1.37, NMV P48873) and similarly proportioned to other described species, G. antarcticus (Ritchie, 1975, fig. 2B) and G. thorezi (Janvier & Clément, 2005, fig. 5B). The shape of the AMV this plate varies between NMV P48874 (Fig. 1) and NMV P48884 (Fig. 12), the caudal portion of the latter being more elongate and closer resembling G. antarcticus in overall shape (Ritchie, 1975, fig. 2B). The posterior median ventral plate (PMV) is trapezoidal and narrow (B/L = 0.53, NMV P48873). The posterior ventrolateral plates (PVL) exhibit a complex form of overlap areas (Fig. 9) characteristic of phlyctaeniid arthrodires (Goujet, 1984).

Pectoral Fin. The right pectoral fin is preserved as articulated dermal scales in the holotype. It is short (33 mm) and broad (47 mm) and covered dorsally and ventrally by small polygonal, non- overlapping scales each covered in short, rounded tubercles (Fig. 1). There is no indication of fin radials, suggesting these elements were not perichondrally ossified. The pectoral fin is seldom fossilized among arthrodires, in the basal arthrodire Sigaspis it is represented by approximately ten ovoid scales near the root of the fin (Goujet, 1973, fig. 1B). In more derived forms it is typically preserved as ossified endoskeletal radialia, e.g., Incisoscutum ritchiei (Dennis & Miles, 1981). The pectoral fin is preserved in outline for Amazichthys, which differs from G. howittensis sp. nov. in being proportionately larger and triangular in form (Jobbins et al., 2022).

Post-thoracic anatomy. The tail of G. howittensis sp. nov. is preserved in lateral aspect in two specimens, the anterior portion in AMF 62537 (Fig. 10), and almost the whole tail posterior to the dorsal and anal fins in NMV P48875 (Fig. 11), only lacking the distal tip of the caudal fin. Both specimens are similarly proportioned based on comparable lengths of the MD (NMV P48875, L = 60 mm and AMF 62537, L = 71 mm). These specimens can provide the first complete restoration of the body shape and squamation for the genus (Fig. 14) and indicates a reconstructed tail length of 158 mm. Based on the length of the MD (60–71 mm) and tail (158 mm) in two specimens, summed with the average length of the skull roof (77 mm) in adult specimens (NMV P48873, AMF 63542, 63535), indicates the total length of G. howittensis sp. nov. is approximately 300 mm. The tail of G. howittensis sp. nov. is relatively stout, roughly 55% of the fish’s total length (Fig. 14) when compared with the groenlandaspidid, Africanaspis (~70% from Gess & Trinajstic fig. 3) and the actinolepid, Bollandapsis (~60% from Schmidt, 1976, p. 6). However, the tail of the former is only known from subadult specimens and might not make a good comparison as armor to tail proportions may change with ontogeny (Ritchie, 2005). The tail proportion of G. howittensis sp. nov. concurs with the overall conserved body-plan of other arthrodires as recently reviewed by Engelman (2024).

Figure 14 G. howittensis sp. nov. reconstruction.

(A) dorsal view. (B) Ventral view. (C) Lateral view, dotted lines indicate overlap regions.

The body scales of G. howittensis sp. nov. display lateral and ventral variation. The lateral tail is covered by sub rhombic, non-overlapping scales 2.5–<0.1 mm in length, each lateral scale exhibits rounded densely packed ornament on their dermal surface, large scales present toward the trunk plates bear a transverse ridge, some of these scales are deeply furrowed by the continuation of lateral canal from the PDL (Fig. 10). A postmedian “scute” (pms) can be observed toward the caudal end of NMV P48875 (Fig. 11), it is similar in morphology to the larger scales toward the base of the tail. Such “scutes” also occur in several other stem gnathostomes, e.g., Kujdanowiaspis and Xuishanosteus (Dupret, 2010; Zhu et al., 2022). Alternatively, the postmedian scute could be a displaced large lateral scale. A portion of the ventral side of the tail is preserved in one specimen, NMV P48884 (Fig. 12), wherein overlapping scales immediately posterior to the base of the PVL plates is transversely elongated and completely lack ornamentation. Our description accords with a previous work describing the lateral body scales in G. howittensis sp. nov. (Burrow & Turner, 1999, fig. 3I) though we describe further variation in the material regarding the ventral scales in NMV P48884 and a possible post median scute in NMV P48875. A possible pelvic girdle is identified from a poorly-defined impression in AMF 62537 (Fig. 10). It shows a slender iliac process (il.proc) and broad basal plate (pelv) as in Gogo arthrodires, e.g., Incisoscutum ritchiei (Dennis & Miles, 1981) though overlying scales obscure finer anatomical detail.

Phylogenetic results

The results of the 50% majority rule tree (Fig. 15) include clades which are identified in the strict consensus of other analyses, e.g., Carr & Hlavin (2010) and Zhu, Zhu & Wang (2015), but are not resolved in our strict consensus due to unstable taxa. A parsimony analysis (heuristic search) of our modified data matrix returned 35234 equally parsimonious trees at 618 steps (Fig. 15). The topology of our 50% consensus analysis is broadly comparable to the strict consensus of Zhu, Zhu & Wang (2015, fig. 9) though we recover lower support values for branches concerning homostiid and dunkleosteid taxa. The two Moroccan eubrachythoracids added in this analysis, Amazichthys and Alienacanthus, emerge as sister taxa nested among other aspinothoracids, in congruence with Jobbins et al. (2024). The node supporting the Brachythoraci is defined by two synapomorphies; a laterally expanded or trapezoidal nuchal plate (char. 105) and contact of the ADL and PL plates (char. 126). The phlyctaeniid node is supported by the following synapomorphies: midline contact of the ADLs (char. 128), an internal thickening of the posterior trunk plates (char. 129) and sigmoidal/double overlapping of the PVL plates (character 130). In the strict consensus groenlandaspidids are nested among the phlyctaeniids, sister to the arctolepids (Heintzosteus and Arctolepis) with Dicksonosteus one node basal. The groenlandaspidid Mulgaspis recovers most basal among groenlandaspidids, followed by Tiaraspis in the 50% consensus. All members of the genus Groenlandaspis, including G howittensis. sp. nov. sit crownward to other groenlandaspidids in our 50% majority rule tree except for Africanaspis which is recovered in a polytomy with G. riniensis basal one node to other species of Groenlandaspis. The incompletely known taxon Elvaspis tuberculata recovers either basal to the phlyctaeniids or basal to the brachythoracids in most parsimonious trees.

Figure 15 50% majority-rule consensus of 35234 equally parsimonious trees showing the phylogenetic relationships of G. howittensis sp. nov. and Groenlandaspididae (highlighted green) among phlyctaenioid arthrodires.

Values at nodes indicate consensus frequency (thus only nodes which occur at 100% will also appear on the strict consensus). Image silhouettes are our own (G. howittensis) or modified from the following: Africanaspis doryssa, (Gess & Trinajstic, 2017, fig. 3); Holonema westolli, (Trinajstic, 1999, fig. 5C); Coccosteus cuspidatus (Trinajstic et al., 2015, fig. 16); Amazichthys trinajsticae (Jobbins et al., 2022, fig. 9).

Discussion

Intraspecific variation

Intraspecies variation is a pervasive problem in the description of fossil organisms. Anatomically distinct specimens can be interpreted as two taxa without the presence of intermediate forms. In some cases the geological distortion at Mount Howitt has affected previous interpretations, Austrophyllolepis youngi, for example, was originally considered distinct from Austrophyllolepis ritchei (Long, 1984). But now considered to result from distortion in the Mount Howitt specimens (Ritchie, 2005). Intraspecific variation, particularly regarding the MD plate has been recognised in other groenlandaspidids, e.g., Mithikaspis, Turrisaspis and Mulgaspis (Young & Goujet, 2003; Daeschler, Frumes & Mullison, 2003; Young & Long, 2014) and some variation is observed in the material of G. howittensis sp. nov. In the material there is variation in the shape of the AMV between NMV P48874 (Fig. 1) and NMV P48884 (Fig. 13), and variation in the presence of the spinelets on the mesial margin of the spinal, absent in the holotype NMV P48873 (Fig. 1), but clearly present in NMV P48875 and NMV P48884 (Figs. 11, 13). Variation in the shape of the AMV has also been shown in extensive material of incisoscutid and camuropiscid arthrodires (Trinajstic & Hazelton, 2007) however, the variation of mesial spinelets to our knowledge is unique to this taxon.

Based on the available material and with comparison of intraspecific variation in other arthrodires, we equate the variance of these morphologies to normal intraspecific variation and we refer the described material to a single species, G. howittensis sp. nov; however, we cannot preclude the existence of two very anatomically close species of Groenlandaspis present in the Mount Howitt fauna.

There is also common asymmetrical variation in the path of sensory canals present on every specimens of G. howittneiss sp. nov. where cranial plates are preserved. For example, on the holotype, the lateral canal (lc) of the right PNu is disjointed and in AMF 63548 (Fig. 5) the left supraorbital canal diverges briefly from its normal path. The most unusual example of this is in AMF 155378 (Fig. 8), where the PNu exhibits a second ‘aberrant canal’ (a.c) which diverges toward the post marginal canal (pmc) and does not readily compare to any sensory canal before described in arthrodires. Asymmetrical variation in the growth of plates and sensory canals in arthrodires has been linked to intense environmental stresses (Trinajstic & Dennis-Bryan, 2009). A similar interpretation has been made of the dipnoan taxa (Barwickia and Howidipterus) of the Mount Howitt site which are thought to have recently diverged from a common ancestor driven by resource scarcity (Long & Clement, 2009).

Comparison of gnathal plates with other arthrodires. In three of four specimens showing the gnathal plates of G. howittensis sp. nov. (NMV P48773, Fig. 2, AMF 62534, Fig. 6, and AMF 62333, Fig. 7) the ASG is preserved under the rostral area with the PSGs angled medially such that they almost connect at the midline. In the fourth specimen (NMV P48884, Fig. 12) the gnathal plates are clearly displaced during deposition, but is congruent with the other specimens in showing only one ASG. Given this arrangement is preserved identically across multiple specimens we hypothesize this reflects their position in-life thus contrasting the generalized arrangement in other arthrodires (Fig. 16). The arrangement hypothesised here can be tested with knowledge of the neurocrania and palatoquadrate to which the ASG and PSG connect respectively but these elements are not preserved in this material. This unusual specialisation has likely led to some error in the interpretation of these elements in other groenlandsaspidids. In Turrisaspis elektor a possible fused ASG is referred to as the ‘anteroventral margin of the rostral plate’ by Daeschler, Frumes & Mullison (2003, fig. 8). A single fused ASG was also noted but not figured by Long et al. (1997, p. 258) in a specimen of a “juvenile G. riniensis”, (subsequently reassigned to Africanaspis doryssa by Gess & Trinajstic, 2017. fig. 2B), but not further described. Both genera appear to show the same unique arrangement of PSG plates as with G. howittensis sp. nov., supporting the likely occurrence of a fused ASG. A review of this material in regard to the gnathals is required, if the presence of a dorsoventrally flattened ASG, as observed in G. howittensis sp. nov., is verified in Turrisaspis and Africanaspis it may be characteristic of the family Groenlandaspididae. The condition of a fused ASG however is not only found in groenlandaspidids however. In non-groenlandapsidid arthrodires a fused ASG, referred to as the ‘medio-gnathal’, is described in the eubrachythoracids Mylostoma (‘Dinognathus’, Dunkle & Bungart, 1945) and Bungartius (Dunkle, 1947) from the Late Devonian, Cleveland Shale, USA. Compared with G. howittensis sp. nov. these forms appear to retain the conventional arrangement, that is, the median ASG is flanked laterally by the PSGs (Hlavin & Boreske, 1973, fig. 2B). A “peg-like” fused ASG was documented for Holonema westolli (Miles, 1971, p. 150) but subsequent newly prepared specimens form Gogo confirm it is a paired element as in other arthrodires (JA Long, 2024, personal observation).

Figure 16 Arrangement of upper-tooth plates in basal arthrodires.

Red = anterior supragnathal (ASG), blue = posterior supragnathal (PSG), green = parasphenoid (Psph). (A) ‘buchanosteid arthrodire’ ANU V244, after Hu, Lu & Young, 2017, fig. 6B. (B) Groenlandaspis howittensis sp. nov. composite reconstruction after NMV P48773 and AMF 62534. (C) Cowralepis mclachlani after Ritchie, 2005, figs. 9F, G & 15C, D. Not to scale.

Functional morphology and paleoecology. The unique arrangement of upper gnathal plates in G. howittensis sp. nov. raises questions about the nature of the jaw occlusion and function of the jaw apparatus. In NMV P48873 the IGs are superimposed onto the PSG bones closely corresponding with the concave surface of the PSG (Fig. 2). We speculate, as shown in this specimen, the lower jaws may have occluded exclusively with the PSGs. This novel adaption might have important implications for the global migration of the family during the Devonian. Nonetheless, without preservation of gut contents or the remaining jaw apparatus (e.g., meckelian cartilage, palatoquadrate, hyoid arch) inferences on the functional significance of this structure or the occlusion of the jaws remain speculative. The ventrally flattened body, dorsolaterally positioned eyes and ventrally positioned mouth, are consistent with bottom feeding habits and a demersal niche characteristic of basal arthrodires (Miles, 1969). A relatively stout, heavily scaled tail suggests G. howittensis sp. nov. was likely a weak swimmer, the short and inflexible pectoral fins likely only assisted in minor lift to keep the fish slightly above the bottom of its lacustrine habitat when it swam. The fine, tuberculate homodont dentition of this species aligns with a villiform morphotype adapted for gripping rather than crushing or puncturing prey common in extant demersal fish, e.g., groupers (Epinephelus Mihalitsis & Bellwood, 2019) or siluriforms (Sado et al., 2020). Alternatively, Gess & Whitfield (2020) interpreted the gnathal plates of G. riniensis as those adapted to a durophages diet, supported by the occurrence of bivalves preserved within some juvenile specimens. A durophagous habit is more likely for those groups living in marine ecosystems whereas this contrasts with the palaeoenvironmental interpretation of the Mount Howitt site as lacustrine, with the only non-vertebrate material identified being lycopsid plants (Long, 1983a). Moreover, the gape of G. howittensis sp. nov. would have been heavily limited by the narrow nuchal gap and extrascapular plates, thus, incapable of feeding on other fully-grown gnathostomes of the Mount Howitt fauna. Though the function of the peculiar dental array cannot be further interpreted at this time, G. howittensis sp. nov. possibly scoured the benthic zone for larval fishes or soft-bodied invertebrates, analogous to extant freshwater skate or catfish.

Systematic implications. G. howittensis sp. nov. is the most completely known described groenlandaspidid and is the first member of the cosmopolitan genus Groenlandaspis to be formally described from Australia. This material reveals new morphologies typically associated with more derived forms, e.g., extrascapular plates and cutaneous sensory pits, which have broad implications for trait acquisition in arthrodires. Extrascapular plates were first considered a specialisation of the brachythoracids (Miles, 1973; Dennis & Miles, 1979; Gardiner & Miles, 1990). But are now known in various, actinolepidids, e.g., Sigaspis, Aleosteus, and Erikaspis (Goujet, 1973; Johnson, Elliott & Wittke, 2000; Dupret, Goujet & Mark-Kurik, 2007), and now the phlyctaeniid, Groenlandaspis, supporting extrascapular elements as being plesiomorphic for arthrodires and so subsequently lost in numerous later groups. But their absence may result from preservation bias. Of several articulated specimens examined for this study only two occurrences of extrascapular plates were identified (AMF 63548, Fig. 5 and AMF 63535 not figured in this article) and one possible extrascapular was identified in disarticulated material (AMF 155378, Fig. 8). King, Hu & Long (2018) reviewed the presence of possible electro sensory organs in Paleozoic gnathostomes. They noted the potential phylogenetic significance of cutaneous sensory pits (char. 126) in arthrodires. This feature is generally restricted to buchanosteids, coccostemorphs as well as Eastmanosteus in our analysis and is clearly present in the phlyctaeniid, G. howittensis sp. nov. (Figs. 4D, 4E). The cheek plates for other groenlandaspidids are poorly known but these elements as described for G. riniensis (Long et al., 1997, fig. 5H) and Africanaspis (Gess & Trinajstic, 2017, figs. 5B, D) show no evidence of sensory pits.

The infraorder Phlyctaenii Miles, 1973 is often considered as a grade group by several workers (e.g., Dennis & Miles, 1979; Gardiner & Miles, 1990, 1994 and Zhu, Zhu & Wang (2015)). Our hypothesis of arthrodire phylogenetic relationships reflects that of Goujet (1984) and Dupret (2004) in supporting a monophyletic relationship of the phlyctaeniid families, Groenlandaspididae, Arctaspididae and Arctolepidae. Another major arthrodire family considered among the Phlyctaenii are the Phlyctaeniidae, Fowler, 1947, (e.g., Phlyctaenius and Pagaeauspis); they lack the unusual overlap pattern of the PVL plates (Young, 1983) and it is unclear if they possess a developed annular bourrelet as in Arctolepis, Dicksonosteus and Groenlandaspis. We propose these forms require further investigation of their phylogenetic relationships, as they are generally considered as a grade group by other workers positioned sister to the rest of Phlyctaenioidei (Goujet, 1984; Dupret, Zhu & Wang, 2017). Our analysis fails to support the monophyly of the genus Groenlandapsis and we do not identify any unique specialisations shared between currently described members of the genus. Though we have provided an amended diagnosis we stress that multiple complete species of Groenlandaspis await further description, namely, G. disjectus from the Kiltorcan Formation, Ireland (Ritchie, 1974), Groenlandaspis sp. from the Adolphspoort Formation, South Africa (Anderson et al., 1999), Groenlandaspis sp. from Canowindra, Australia and an abundance of fragmentary material from multiple other sites in Australia (Young, 1993). As such, our diagnosis for Groenlandaspis should be considered tentative. Revision of the type species G. mirabilis is also necessary as some bones remain misidentified, e.g., the “AMV” and “AVL” only depicted by drawings in Heintz (1932, fig. 12) differ strongly in shape from any known arthrodires and are likely erroneously labelled PVL plates. A full taxonomic review of Groenlandaspis is required to complete a definition of the genus and further probe its phylogenetic relationships. Gess & Trinajstic (2017) discussed affinities of the “high-spired” groenlandaspidids, Tiaraspis, Turrisaspis and Africanaspis primarily the presence of a dorsolateral ridge, dual pineal elements, and the foreshortened trunk armour, compared with most species of Groenlandaspis. Dual pineal elements (char. 122) are now properly described in Groenlandaspis and this is likely a synapomorphy uniting a clade of derived groenlandaspidids, with a single element exhibited by Arctolepis and Mulgaspis being the plesiomorphic state. Likewise, a dorsolateral ridge (char. 126) is not restricted to “high-spired” genera, and is commonly reported among phlyctaeniid taxa, e.g., Denisonosteus (Young & Gorter, 1981) and Phlyctaenius (Young, 1983), though lost in Mulgaspis and some species of Groenlandaspis and thus cannot be characteristic of “high-spired” genera. Lastly, compared to Groenlandaspis, the trunk armour of Turrisaspis and Africanaspis and to a lesser extent Tiaraspis are foreshortened in proportions, particularly in the median dorsal plate (Long et al., 1997; Daeschler, Frumes & Mullison, 2003). Though similarly foreshortening is present in some Groenlandaspis species, as in the ADL and PDL of G. riniensis (Long et al., 1997, figs. 7A, B) and the MD of G. seni (Janvier & Ritchie, 1977, figs. 1B, C). The significance of this morphology requires further investigation to quantify the effect of bone proportions on the phylogeny of groenlandaspidids. The phylogenetic relationships of these “high-spired” groenlandaspidids naturally relies on the completeness of their record. Material from the Middle Devonian Aztec Fauna, Antarctica consisting of an incomplete tall MD plate referred to the genus Boomeraspis as well as isolated ADL and PDL plates show a general resemblance to the Late Devonian Turrisaspis. (Young & Long, 2014). Likewise, the Emsian genus Mithikaspis may also be related to Tiaraspis based on the height of the MD (Young & Goujet, 2003). These taxa are temporally intermediate between the Early Devonian Tiaraspis (Schultze, 1984) and Late Devonian Turrisaspis and Africanaspis (Daeschler, Frumes & Mullison, 2003; Gess & Trinajstic, 2017), indicating further sampling of Early to Middle Devonian groenlandaspidids is could elucidate these relationships further.

Conclusion

G. howittensis sp. nov. provides us with rare insight into the morphology of the post-trunk skeleton, fins and dental morphology for arthrodires. The exceptional preservation of the Mount Howitt specimens reveals new details of the gnathal plates for groenlandaspidids, thus adding valuable information to our knowledge of gnathal plates in basal arthrodires and highlights a uniquely specialised condition where the ASG is fused and positioned anterior to the remainder of the gnathal arcade. G. howittensis sp. nov. is a unique example of extreme dental specialisation and evolutionary experimentation in stem jawed vertebrates. The phylogenetic relationships of the Groenlandaspididae are published for the first time in a computer-driven phylogenetic analysis and supports a position among basal arthrodires but the interrelationships of groenlandaspidid genera require further investigation.

Supplemental Information

Supplemental Information 1 PAUP output generated after heuristic search.

PAUP generated two trees. First tree = strict consensus, second tree = 50% majority consensus.

Supplemental Information 2 Phylogenetic matrix modified after Zhu et al. (2016).

Supplemental Information 3 Corrections made of the phylogenetic scores used in Zhu et al. (2016).

Supplemental Information 4 Full phylogenetic character list.

We are grateful to Dr. Matthew McCurry of the Australian Museum for graciously making latex peels of many specimens in their collection. We thank Tim Ziegler for providing access to the palaeontological collections of the Melbourne Museum and for his assistance in locating specimens. We thank Shona Ritchie and the Canowindra Age of Fishes Museum for access to the notes and casts of specimens made by the late Dr. Alex Ritchie. We would like to thank Russel Engelman and Sebastien Olive who made comments on the preprint which improved the manuscript. We appreciate the feedback of the reviewers, Melina Jobbins and two anonymous, who gave extensive comments which greatly improved the manuscript.

Institutional abbreviations

NMV Museum of Victoria, Melbourne, Australia

AMF Australian Museum, Sydney, Australia

ANU Australian National University, Canberra, Australia

Anatomical abbreviations

ab annular bourrelet

a.c aberrant canal

acc accessory canal

ADL anterior dorsolateral plate

af anal fin

AL anterior lateral plate

AMV anterior median ventral plate

APi anterior pineal plate

ASG anterior supragnathal

AVL anterior ventrolateralplate

C central plate

cf.ADL contact face for the anterior dorsolateral plate

cf.AMV contact face for the anterior median ventral plate

cf,MD contact face for the median dorsal plate

cf.PDL contact face for the posterior dorsolateral plate

cf.IL contact face for the interolateral plate

cf.PMV contact face for the posterior median ventral plate

cf.PVL contact face for the posterior ventrolateral plate

cf.Sp contact face for the spinal plate

csc central sensory canal

cr.PNu paranuchal crista

cuso cutaneous sensory organ

df dorsal fin

end.d endolymphatic duct

Esc extrascapular plates

Esc.c extracapsular plate canal

if.pt infranuchal pit

IG infragnathal

IL interolateral plate

il.proc iliac process of the pelvic gridle

ioc infraorbital canal

kd articular condyle

lc lateral canal

l.infsp infraspinal lamina

MD median dorsal plate

mpl median pit line

Nu nuchal plate

oa.AVL overlap area for the anterior ventrolateral plate

oa,C overlap area for the central plate

oa.IL overlap area for interolateral plate

oa.M overlap area for the marginal plate

oa.N overlap area for the nuchal plate

oa.PL overlap area for the posterior lateral

oa.PVL overlap area for the posterior ventrolateral plate

occ occipital cross commissure

orb orbit

pap para-articular process

PDL posterior dorsolateral plate

pdl posterior descending lamina

pect.f pectoral fin

pelv basal plate of the pelvic girdle

pelv.f pelvic fin

PL posterior lateral plate

PPi posterior pineal plate

ppl posterior pit line

ppt pineal pit

psoc post suborbital canal

PM post marginal plate

pmc postmarginal canal

pms post median scute

PMV posterior median ventral plate

PNu paranuchal plate

PrO preorbital plate

PSG posterior supragnathal

PSO post suborbital plate

Psph parasphenoid

PtO postorbital plate

PVL posterior ventrolateral plate

R rostral plate

SM submarginal

SO suborbital

soc supraorbital canal

sorc supraoral canal

Sp spinal plate

suo.v supra orbital vault

symph.s symphysial surface

v.gr ventral groove

Additional Information and Declarations

Competing Interests

Author Contributions

Data Availability

New Species Registration

The authors declare that they have no competing interests.

Austin N. Fitzpatrick conceived and designed the experiments, performed the experiments, analyzed the data, prepared figures and/or tables, authored or reviewed drafts of the article, and approved the final draft.

Alice M. Clement analyzed the data, authored or reviewed drafts of the article, and approved the final draft.

John A. Long conceived and designed the experiments, analyzed the data, authored or reviewed drafts of the article, and approved the final draft.

The following information was supplied regarding data availability:

The character matrix used for the phylogenetic analysis in this study is available in File S2.

The following information was supplied regarding the registration of a newly described species:

Publication LSID: urn:lsid:zoobank.org:pub:42499E77-9848-4655-A606-589C30F41D72

Groenlandaspis howittensis LSID: urn:lsid:zoobank.org:act:179A4F7E-13D6-49AB-ADBE-9545664E50E1.

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
