# Peer review of "Unique dental arrangement in a new species, Groenlandaspis howittensis (Placodermi, Arthrodira) from the Middle Devonian of Mount Howitt, Victoria, Australia"

_PeerJ, doi:10.7717/peerj.18759_

## Round 0.1 · original submission · Major Revisions

This manuscript has been reviewed by three experts in the field. All of them believed it is worthy to be published, and in fact all 3 have opted for "Minor Revisions". However, they also raised some concerns and suggestions on the current form of the manuscript, which is essential for furthering improving it. There are still some inconsistencies mentioned by reviewers, the authors need to follow the journal criteria or those recent-related published papers.

·

Basic reporting

No comment

Experimental design

No comment

Validity of the findings

No comment

Additional comments

This is a great paper introducing a new arthrodire species and reassessing the monophyly vs paraphyly of Groenlandaspis. The specimens do not disappoint in quality and provide a great amount of information, all well detailed by the authors. There are very minor punctuation issues in a few places (marked in red in the ms) and a few notes regarding some figures:
- Fig 1 B and C are reversed when reading the caption
- Fig 3 A line drawing would make a nice addition. Also, the posterior end of the Nu is not in focus, a stacked photo would solve this.
- Fig 5 is a little dark to see some of the structures like if.pt
- A map showing the location of the site in Australia would be a great add as well

Reviewer 2 ·

Basic reporting

The manuscript is, overall, well written; I have marked suggestions for minor grammatical/spelling corrections on the annotated pdf – e.g., inappropriate use of commas. The introduction could be expanded to mention the ‘controversy’ over placoderm monophyly vs polyphyly as background information, and to include more of the historical information on studies of groenlandaspidids. The literature is well referenced and relevant; I have added some corrections or highlighted items that need attention in the annotated pdf. As one of the main features of the work is the phylogenetic analysis, I believe the authors should add mention of the unpublished (but available via doi:10.13140/RG.2.2.24314.98249) PhD work of Cynthia Deschênes which included a phylogenetic analysis of groenlandaspidids.
I would hope that editors will assess whether the format of the manuscript fits the Journal criteria; I started marking some inconsistencies on the pdf (e.g. presence/absence of , after authors’ names in text references). I have not checked whether the reference list is complete, as I assume editors will handle this. A synonymy should be added for the new species, which should also be registered with Zoobank. I agree that the taxon is a new species, and is correctly described cf. ICZN standards.
Figures are good quality, mostly well labelled; captions could perhaps be expanded.

Experimental design

The work is original as far as I can assess it, and technically sound. Authors have clearly stated the aims and results of their project, showing how the work extends knowledge of this group of vertebrates.

Validity of the findings

Data matrix for the phylogenetic analysis needs to have details of ‘pre-used’ characters added, or else a suppl word document listing them, rather than expecting readers to work them out from two or three or more other publications. The actual analysis is fine, performed as per commonly used programs. The Conclusion section adequately covers the results of the research, with some clarification as requested on the annotated pdf.

Additional comments

The authors have given an extensive description and illustration of this new species, comparing it with many related taxa. These comparisons could be expanded, as I have noted on the annotated pdf.
One issue that has been glossed over is the presence/absence of real teeth in this (and related) fish. Clearly as the work is based on casts of specimens, the histology of its hard tissues is not known, but the authors should provide some justification for regarding them as teeth – have real teeth been identified in any groenlandaspids (or even phlyctaenioids)? As far as I recall, Smith & Johanson (2003, suppl www.sciencemag.org/cgi/content/full/299/5610/1235/ ) described teeth in an actinolepidid and eubrachythoracids, but noted other arthrodire clades appeared to lack teeth.

Annotated reviews are not available for download in order to protect the identity of reviewers who chose to remain anonymous.

Reviewer 3 ·

Basic reporting

This manuscript is presented in clear English language, the Introduction, Structure and Figures are adequate, and raw data is supplied. However, specific comments by line number [provided separately] suggest various amendments to the text, and some of the figures.
The literature is well-referenced, however the following seem relevant to the topic, but have been overlooked. These could be included in the reference list, and cited at appropriate places if relevant, to be incorporated in a revision of the text.
Young G.C. [2003]. Did placoderm fish have teeth? Journal of Vertebrate Paleontology 23: 987-990.
Young G.C. & Long J.A. [2014]. New arthrodires (placoderm fishes) from the Aztec Siltstone (late Middle Devonian) of southern Victoria Land, Antarctica. Australian Journal of Zoology 62, 44-62.
Burrow, C., Hu, Y., Young, G. [2016]. Placoderms and the evolutionary origin of teeth: a comment on Rucklin & Donoghue (2015). Biology Letters 12, 20160159 (http://dx.doi.org/10.1098/rsbl.2016.0159). 

Hu, YZ, Lu, J & Young G.C. [2017]. New findings in a 400 million-year-old Devonian placoderm shed light on jaw structure and function in basal gnathostomes. Scientific Reports 7: 7813. doi:10.1038/s41598-017-07674-y
Hu YZ, Young, GC, Burrow, C, Zhu, YA, Lu, J, [2019]. High resolution XCT scanning reveals complex morphology of gnathal elements in an Early Devonian arthrodire. Palaeoworld https://doi.org/10.1016/j.palwor.2018.12.003

Experimental design

Original research is within the scope of the journal, the research question is well-defined, and information is presented in sufficient detail for replication. Again, specific comments by line number [provided separately] propose some amendments to these aspects.

Validity of the findings

The conclusions require clarification and rewording, as noted by specific comments by line number [provided separately]. There are various generalised statements of the type that might be made to conclude a description of routine fossil material. However the specimens on which this new species is based have exceptional preservation, providing new information that may be extrapolated to other genera, or even higher taxa. That should be the emphasis of this paper.

Additional comments

This manuscript is suitable for publication, subject to revision. It describes and illustrates very significant new material. The description and illustrations are generally well-structured, but additions or clarifications can be proposed. A problem in some parts of the descriptions are vague statements regarding which specimens demonstrate which morphological structures [e.g. ‘in one/several specimens’]. Please be specific. This is only a small collection, so specimen numbers should replace the vague statements. In addition, for described many structures, it needs to be clarified how many of the specimens show it, whether preserved on one or both sides, or missing from the specimen, etc. This is crucial information regarding evidence for intraspecific variation.

Detailed comments by line number [provided separately] make specific all these suggestions. All listed comments are also marked on the attached PDF, together with numerous minor typographic corrections.
Specific Comments on the text [by line number]
Line 2. Suggested amendment to the title brings it in line with ‘new species policies’ for PeerJ.
Line 21. I suggest ‘a new species’ [no longer ‘undescribed’ when this manuscript is published].
Line 39. This is controversial -- see Young [2003] and Burrow et al. [2016].
Lines 65-67. This seems somewhat superfluous to the ‘Introduction’. Dispersal patterns cannot be ‘demonstrated’ [perhaps hypothesised], and certainly nothing is demonstrated by the cited Dupret et al. paper [except perhaps incompetence, in my view].
Line 78. Surely Abbreviations [‘Material’] should come before phylogenetic analysis [‘Methods’].
Lines 80-81. Superfluous -- suggest delete.
Lines 142-43. The new generic diagnosis presented here assumes an unpaired ASG occurs in Groenlandaspis mirabilis and other species, but not in other groenlandaspids. This assumption should be stated [with evidence presented elsewhere; see below].
Lines 150-1. This could be clarified; not exactly what ‘researchers’ said. As Janvier & Clement 2005 explained, a monophyletic genus would be straightforward, except that other groenlandaspids are now named, and included in a monophyletic family diagnosed by Young & Goujet 2003. The problem then is to distinguish Groenlandaspis from other genera, made more difficult because key features [e.g. MD plate, symphysis of ADL, PDL plates] are unknown in the genotype Groenlandaspis mirabilis.
Line 154. A synonymy list should be included in the formal erection of a new species.
Lines 161-64. This seems odd. There are two pieces of rock with fossils preserved, but they represent only one fish. Surely this fish is the holotype. The various other specimens illustrated in the figures are the referred material. These require a full list under that heading.
Line 174. If there are ‘several specimens’ -- how many? But none are listed under ‘Referred Material’ on the previous page.
Line 176-7. The supraorbital sensory canal is restored as deflected laterally at the anterior end [Figure 14A] compared with the mesial deflection in other forms [e.g. Arctolepis, Mulgaspis]. This should be noted/discussed. Is this feature variable within the material?
Line 179-180. But in Groenlandaspis and Mulgaspis the PrO forms the pre-orbital corner. This is the essential difference.
Line 184. Does this mean it is variable within the new species? Surely the pineal opening should be mentioned -- a key structure for this bone.
Line 189. This is an interesting proposal – but if correct, according to Schultze 1984 the pineal opening is on the posterior plate, not the anterior as described here. This should be commented on.
Line 194. Does this mean ‘deeper ebayment’? The Figure 14 reconstruction shows a transverse anterior margin, compared with the deep indentation between sensory grooves for the Antarctic Groenlandaspis. Is this a difference? Are these features variable in the several skulls of the new species?
Line 204. Suggest ‘and may have superficially connected with …’, if that wording represents the interpretation.
Line 209-224. Clarify first which specimens have the cheek preserved. Is the description based only on the holotype?
Line 220. Give the specimen number and fig. number first. ‘near complete’ or ‘complete’? Fig. 6 suggests its posterior edge is obscured [and SM looks a bit like SO on the opposite side]. More detail would be useful. Given this is claimed the first SM described for a Groenlandaspis, the description should be expanded and clarified.
Line 229. Clarify. As worded it implies brachythoracids don’t show wear facets [wrong]. Dunkleosteus has blade-like tooth plates, very different from these denticulate bones. Hu et al. (2017, 2019) give details of the primitive brachythoracid condition [denticulate] for the three gnathal elements. These papers should be considered and cited. Are denticles pointed or rounded in the new material? The latter could indicate wear.
Line 242. Misleading. The pineal organ is at the top of the braincase, and the PSG at the base, but the braincase is not preserved. So this is only a preservational position.
Line 248. If so, comment on ASG in these other forms; e.g. do specimens preserve the relevant area? Have they fallen off? And why is it assumed the ASG is fused in the others [as implied by the generic diagnosis]?
Line 251. The Young et al. [2001] description is completely superseded by CT scanning and 3-D printing, as described by Hu et al. [2017, 2019].
Line 255-56. Fig. 5 suggests two divisions i.e. larger, denser denticles only on the anterior [mesial] half. The Stensio 1963 reference is irrelevant [cf. Hu et al. 2017, 2019]. Ossification centre position, present/absence of the non-biting division etc. were discussed as characters for phylogenetic analysis by Young et al. [2001, pp. 676-77], descriptions updated by Hu et al. [2017, 2019]. This could be considered [also in the ‘Phylogenetic Analysis’ section]. A good drawing of the Howitt IG would be very useful.
Line 263. Again, please state for each plate how many preserved examples [i.e. can intraspecific variation be demonstrated?]
Line 265. Is the symphysis with the opposite ADL preserved? Add to description.
Line 266. Clarify. If only one example is sufficiently complete to be measured, state this please.
Line 265-76. Do these general statements add much? At least move to after the description of the various bones.
Line 277. Why only broken in adults, and how to distinguish adult vs juvenile? What is the evidence? But this is a separate topic – please focus here on describing the MD – how many examples, etc.?
Line 280. Clarify. If only one example is sufficiently complete to be measured, state this please.
Line 280-81. Clarify. Do you mean just this specimen? Do others show it?
Line 291. I don’t understand. If they both have overlaps, they had a common suture, thus excluding midline contact of AVLs [as shown in Fig. 14B].
Line 297-301. This could be clarified. If the fin was naked, radialia would be seen. What other taxa have the fin covered in scales, as here?
Line 308. Clarify. The first for the genus, or the first for any groenlandaspid?
Line 312-14. But this ‘tilt’ is included in the reconstruction [Figure 14C], so that would give a reasonable length estimate. Rewording required.
Line 316-17. This could be clarified – is the tail more stout than the others, or greater or less than half total length? Sigaspis and Bollandaspis Schmidt 1976 are close relatives with the tail preserved [I think the latter much more elongate], but not mentioned. Some comparative comments could be made about them.
Line 319-23. Clarify. Did Burrow & Turner illustrate them? Most important is to present proper descriptions of the scales based on these new specimens. Describe first, and if in agreement with the earlier description [or otherwise], then state that.
Line 368. MD variation is discussed in detail by Young & Long [2014, pp. 55–60], including new Antarctic groenlandaspid material [MD of Boomeraspis, ?Tiaraspis, ?Mulgaspis]. This paper is not in the references. Given that the Howitt and Antarctic regions were together in the Devonian, it should be discussed and cited.
Line 369-73. This information should be with the description of these elements, not in the discussion section.
Line 375-78. I suggest to reword this section as a new paragraph:
‘Based on available material, and comparison with intraspecific variation in other forms, all specimens described here are referred to one species Groenlandaspis howittensis. New material from the Howitt locality could change this interpretation.
Line 387. Reword? ‘A similar interpretation has been made ….’
Line 390-93. Evidence for this statement? The following arguments [lines 394–404] are jumbled and need rewriting. The fused ASG is demonstrated in the new species. State evidence where it is seen in other Groenlandaspis species, and other groenlandaspid genera. Where not preserved, clarify this. It is only an assumption that this specialisation applies to the family [if that is being proposed].
Line 405-410. I think this is complete speculation. No evidence is presented regarding occlusion between lower and upper jaws, and this is unknowable without the jaw cartilages preserved [see Hu et al. 2017]. Comparing upper and lower jaw denticles might indicate something.
Line 421-22. Justify this statement.
Line 443-49. This seems either irrelevant, or too vague – were sensory pits described or figured for this new species?
Line 450-501. This is rambling and repetitive -- most could be removed in my opinion. The phylogenetic analysis parts should be condensed into the ‘Phylogenetic Results’ section [lines 334–356].

Specific Comments on the figures
Fig. 1. B, C letters to be reversed to make consistent with the caption.
Fig. 3. A good tracing of the skull AMF 63548 showing bone sutures would be very helpful.
Fig. 5. A good drawing of the IG would be very useful.
Fig. 14. ‘howittensis’ misspelt in caption.
Fig. 16. ‘Buchanostiid arthrodire’ source should be cited in caption. Fig. 16A should be updated from Hu et al. [2017].

Annotated reviews are not available for download in order to protect the identity of reviewers who chose to remain anonymous.

---

## Round 0.2 · accepted · Accept

The manuscript has been further revised based on feedback from three experts and is now deemed suitable for publication.

Reviewer 2 ·

Basic reporting

The authors appear to have addressed all the issues raised in the first round of review. Some minor grammatical errors are present, but I assume the editors will sort these out.

Experimental design

no comment

Validity of the findings

no comment